# Notoginsenoside R1 Promotes Migration, Adhesin, Spreading, and Osteogenic Differentiation of Human Adipose Tissue-Derived Mesenchymal Stromal Cells

**DOI:** 10.3390/molecules27113403

**Published:** 2022-05-25

**Authors:** Haiyan Wang, Yongyong Yan, Haifeng Lan, Nan Wei, Zhichao Zheng, Lihong Wu, Richard T. Jaspers, Gang Wu, Janak L. Pathak

**Affiliations:** 1Affiliated Stomatology Hospital of Guangzhou Medical University, Guangdong Engineering Research Center of Oral Restoration and Reconstruction, Guangzhou Key Laboratory of Basic and Applied Research of Oral Regenerative Medicine, Guangzhou 510182, China; wanghaiyan@gzhmu.edu.cn (H.W.); yanyongyong@gzhmu.edu.cn (Y.Y.); weinan365@126.com (N.W.); zhichaozheng@aliyun.com (Z.Z.); wcanhong@163.com (L.W.); 2Laboratory for Myology, Department of Human Movement Sciences, Faculty of Behavioural and Movement Sciences, Vrije Universiteit Amsterdam, Amsterdam Movement Sciences, 1081 BT Amsterdam, The Netherlands; 3Department of Orthopaedic Surgery, The Third Affiliated Hospital of Guangzhou Medical University, Guangzhou 510150, China; blueseaww@163.com; 4Department of Oral and Maxillofacial Surgery/Pathology, Amsterdam UMC/Vumc Location and Academic Centre for Dentistry Amsterdam (ACTA), Vrije Universiteit Amsterdam, Amsterdam Movement Science, 1081 HV Amsterdam, The Netherlands; 5Department of Oral Cell Biology, Academic Centre of Dentistry Amsterdam (ACTA), University van Amsterdam and Vrije Universiteit Amsterdam, 1081 LA Amsterdam, The Netherlands

**Keywords:** Notoginsenoside R1, osteoblast differentiation, adhesion, migration, immunomodulation

## Abstract

Cellular activities, such as attachment, spreading, proliferation, migration, and differentiation are indispensable for the success of bone tissue engineering. Mesenchymal stromal cells (MSCs) are the key precursor cells to regenerate bone. Bioactive compounds from natural products had shown bone regenerative potential. Notoginsenoside R1 (NGR1) is a primary bioactive natural compound that regulates various biological activities, including cardiovascular protection, neuro-protection, and anti-cancer effects. However, the effect of NGR1 on migration, adhesion, spreading, and osteogenic differentiation of MSCs required for bone tissue engineering application has not been tested properly. In this study, we aimed to analyze the effect of NGR1 on the cellular activities of MSCs. Since human adipose-derived stromal cells (hASCs) are commonly used MSCs for bone tissue engineering, we used hASCs as a model of MSCs. The optimal concentration of 0.05 μg/mL NGR1 was biocompatible and promoted migration and osteogenic differentiation of hASCs. Pro-angiogenic factor VEGF expression was upregulated in NGR1-treated hASCs. NGR1 enhanced the adhesion and spreading of hASCs on the bio-inert glass surface. NGR1 robustly promoted hASCs adhesion and survival in 3D-printed TCP scaffold both in vitro and in vivo. NGR1 mitigated LPS-induced expression of inflammatory markers IL-1β, IL-6, and TNF-α in hASCs as well as inhibited the RANKL/OPG expression ratio. In conclusion, the biocompatible NGR1 promoted the migration, adhesion, spreading, osteogenic differentiation, and anti-inflammatory properties of hASCs.

## 1. Introduction

With the prolongation of life span and the aging of the population, diseases such as tumors and osteoporosis are increasing rapidly. In addition, cases of bone defects caused by trauma, tumors, infections, congenital dysplasia, or developmental malformations are also increasing [1,2,3]. Osteoporotic fractures are extremely common in the United States, with an estimated 1.5 million fragility fractures each year. In the UK, one in two women and one in five men over the age of 50 years may suffer an osteoporotic fracture in their lifetime [4]. More than 2 million bone defect repair surgery are performed worldwide annually [5]. The repair of bone defects has become a major clinical challenge. In recent years, tissue engineering technology has become a hotspot in the field of bone defect repair. Bone tissue engineering usually uses biological scaffolds with the addition of seed cells and growth factors to promote bone regeneration [6,7].

Bone morphogenetic proteins (BMP), especially BMP-2 and BMP-7, are the most extensively used agents to promote bone regeneration [8,9]. However, BMPs have the disadvantages of high cost and short half-life in clinical use [10,11]. In addition, it may also lead to several potential side effects on other tissues and organs, such as excessive osteogenesis or ectopic bone formation [12,13]. Therefore, the search and development of alternative bone regenerative agents have been receiving more and more attention. Chinese herbology, as an important part of traditional medicine in China, has the advantages of a wide range of sources, good biocompatibility, low toxicity, and no or few side effects. Natural small molecule compounds can trigger certain cellular responses through signaling cascades which can exert anti-tumor [14], anti-oxidant [15], anti-bacterial [16], anti-inflammatory [17], and pro-osteogenesis [18] effects. These compounds can be linked to the scaffold material as a substitute for growth factors to promote bone regeneration [19,20]. *Panax notoginseng* is a traditional Chinese medicine with a long history, which is used as a tonic and hemostatic drug [21]. Total *Panax notoginseng* saponin (PNS) has been reported to prevent bone loss and promote osteogenic differentiation [22]. Notoginsenoside R1 (NGR1) is a primary bioactive natural compound of PNS [23]. Previous studies have reported many biological activities of NGR1, including cardiovascular protection [24,25], neuro-protection [26,27] and anti-cancer effects [28,29]. Recently, it has been demonstrated that NGR1 dose-dependently promotes osteogenic differentiation of MC3T3-E1 pre-osteoblasts [22,30,31]. Mesenchymal stromal cells (MSC) are the most commonly used seed cells in bone tissue engineering [32,33,34,35]. Compared to human bone marrow-derived MSCs (BMSCs), human adipose tissue-derived MSCs (hASCs) have promising potential as seed cells for bone tissue engineering, due to their easy accessibility, high yield efficiency [36], and low donor-site morbidity [37,38,39]. However, little is known about the effect of NGR1 on biological activities and osteogenic differentiation of hASCs required for bone tissue engineering application.

The success of bone tissue engineering mainly depends on cell migration, adhesion, spreading, osteogenic differentiation, and in vivo survival of precursor cells [40]. Doping of bioactive agents on scaffold surfaces had been reported to enhance these biological properties of scaffolds. Therefore, it is highly relevant to analyze the effect of NGR1 on migration, adhesion, spreading, and osteogenic differentiation of precursor cells.

During normal bone regeneration, acute inflammation begins after injury and is resolved almost immediately to ensure normal tissue repair and bone formation [41]. Acute inflammation is necessary for bone healing after injury. However, prolonged inflammation may lead to poor bone healing by altering the balance of inflammatory cells and inflammatory cytokines. Additionally, chronic inflammation leads to the differentiation and activation of osteoclasts, which is detrimental to the healing of bone defects [42]. This suggests that potential biological strategies to promote bone formation might entail relieving chronic inflammation. MSCs can potentially be used for cell-based therapy for inflammation, attributed to their ability to modulate the immune system and secrete important bioactive factors. MSCs are capable of migrating to inflammatory sites and exerting anti-inflammatory effects [43]. Insight into the anti-inflammatory effects of NGR1 of MSCs may be relevant for bone tissue regeneration in inflammatory conditions.

This study aimed to investigate the effects of NGR1 on hASCs’ fate and function, including cell adhesion, cell proliferation, cell migration, and osteogenic differentiation. We also investigated the anti-inflammatory effect of NGR1 in hASCs. Our results indicated that NGR1 at a concentration of 0.05 μg/mL is biocompatible to hASCs and promotes migration, adhesion, spreading, osteogenic differentiation, and anti-inflammatory properties of hASCs.

## 2. Results

### 2.1. Biocompatible NGR1 Enhances Osteogenic Differentiation of hASCs

Ideal bone regenerative therapeutic agents should be biocompatible. In this study, NGR1 concentration up to 5 μg/mL did not inhibit the viability of hASCs on days 1, 4, and 7 (Figure 1A). However, NGR1 promoted osteogenic differentiation of hASCs. ALP is an early-stage osteogenic differentiation marker. On the fourth and seventh day of treatment, NGR1 at a concentration of 0.05 μg/mL upregulated ALP expression in hASCs (Figure 1B,C). This result was further supported by the ALP staining study (Figure 1D). Osteocalcin (OCN) is a marker of late osteogenic differentiation marker. On the 14th day of treatment, NGR1 (0.05 μg/mL) showed the highest stimulatory effect on OCN expression in hASCs (Figure 1E). We further confirmed the osteoinductive potential of NGR1 by analyzing the matrix mineralization in hASCs culture (Figure 2). All the tested concentrations of NGR1 promoted matrix mineralization on days 21 and 28 of the culture (Figure 2A–C). NGR1 at 0.05 μg/mL concentration showed a higher trend of effect on matrix mineralization in the hASCs culture compared with 0.01 and 0.5 μg/mL. NGR1 (0.05 μg/mL) enhanced the expression of early and late osteogenic markers ALP, COL1A1, and OCN and as well as the expression of osteogenic and pro-proangiogenic vascular endothelial growth factor (VEGF) (Figure 3A–D).

### 2.2. NGR1 Increases the Adhesion, Spreading, and Migration of hASCs

More hASCs adhered with higher spreading in the NGR1-treated group compared to the control group (Figure 4A). Quantification of cell adhesion indicated ~1.4 times as many cells attached in the coverslip in presence of NGR1 (Figure 4B). Similarly, cell area was significantly higher in the NGR1 treated group at 4 and 24 h compared to that of the control group (Figure 4C).

To determine the effect of NGR1 on the migration of hASCs, a transwell migration assay and scratch wound assay were performed. The transwell migration assay indicated that NGR1 at a concentration of 0.05 μg/mL robustly promoted hASCs migration compared to at 5 μg/mL (Figure 5A,C). A similar result was further confirmed by the scratch wound assay (Figure 5B,D).

We further analyzed the effect of NGR1 on cell adhesion on 3D-printed TCP scaffolds. Significantly more numbers of the cell were observed on the TCP scaffold in presence of NGR1 (0.05 μg/mL) at 1.5, 4, and 24 h (Figure 6A–C). Quantification of adhered cells in TCP scaffold revealed 2.92-, 2.48-, and 3.31-fold higher cell numbers in presence of NGR1 at 1.5, 4, and 24 h, respectively (Figure 6D–F).

### 2.3. NGR1 Promotes hASCs Adhesion and Survival in 3D Printed TCP Scaffolds In Vivo

Cell-scaffold constructs implanted subcutaneously in nude mice were imaged up to 10 days after implantation. At the subcutaneous implantation site, viable grafts were detected, showing bright fluorescent spots (Figure 7A). The fluorescence intensity of hASCs was reduced over time. At all the time points, the intensity of fluorescence signals in the NGR1 group was higher than in the control group (Figure 7B), which indicates that NGR1 enhanced the adhesion and in vivo survival rate of hASCs on β-TCP scaffolds.

### 2.4. Anti-Inflammatory Effects of NGR1 in hASCs

LPS treatment was used to induce inflammation in hASCs. The anti-inflammatory effect of low and high doses of NGR1 in LPS-treated hASCs was further evaluated by analyzing the expression of inflammation markers IL-1β, IL-6, and TNF-α (Figure 8A–C). LPS treatment enhanced the gene expression of IL-1β, IL-6, and TNF-α in hASCs. A low dose of NGR1 alleviated the LPS-induced expression of IL-1β and TNF-α. Additionally, a high dose of NGR1 alleviated the LPS-induced expression of anti- IL-1β and IL-6 in hASCs. LPS treatment has no significant effect on the gene expression of anti-inflammation markers IL-10 and TGF-β (Figure 8D,E). However, a high dose of NGR1 reduced TGF-β expression relative to the LPS group. A higher level of receptor activator of nuclear factor the kappa B ligand (RANKL)/OPG expression ratio in osteoblast lineage cells results in osteoclast formation and osteoclastic bone resorption. LPS treatment enhanced RANKL expression but did not affect the OPG expression and RANKL/OPG expression ratio in hASCs (Figure 8F–H). Interestingly, NGR1 alleviated the LPS-induced RANKL expression. Only high dose NGR1 alleviated the OPG expression in hASCs. The low dose of NGR1 showed prominent inhibition of the RANKL/OPG expression ratio in hASCs.

## 3. Discussion

Bone tissue engineering technology has developed rapidly and has great potential in biomedical applications. Osteointegration and osteoinductive properties of materials determine their efficacy for bone regeneration. The key cells’ cellular activities, such as adhesion, spreading, proliferation, migration, and differentiation, are indispensable for the success of bone tissue engineering. NGR1, one of the main bioactive compounds from *Panax notoginseng* root, has been reported to promote proliferation and osteogenic differentiation of MC3T3-E1 pre-osteoblasts [30]. In this study, NGR1 did not inhibit the proliferation of hASCs. NGR1 significantly increased adhesion, spreading, and migration of hASCs. NGR1-treated hASCs showed higher adhesion in 3D-printed TCP scaffold in vitro and better survival in vivo. In addition, NGR1 promoted osteogenic differentiation of hASCs as indicated by higher ALP activity, OCN expression, matrix mineralization, and expression of osteogenic differentiation markers ALP, COL1A1, and OCN. NGR1 mitigated LPS-induced expression of inflammatory markers IL-1β, IL-6, and TNF-α in hASCs as well as the RANKL/OPG expression ratio. In summary, the biocompatible NGR1 has the potential to induce migration, adhesion, and osteogenic differentiation of precursor cells, as well as an anti-inflammatory effect (Figure 9).

NGR1 promotes osteoblast differentiation, but the optimal concentration of NGR1 treatment remains debatable. Liu et al. reported that treatment with 5–1000 µg/mL NGR1 promoted osteogenic differentiation of MC3T3-E1, while NGR1 at 200 and 1000 µg/mL significantly inhibited cell proliferation [30]. Huang et al. reported that NGR1 at a concentration of ≤20 µM could promote differentiation of human alveolar osteoblasts (HAOBs) in a TNF-α induced inflammatory microenvironment [44]. NGR1 at a concentration of ≤ 20 µM stimulates rat osteoblast proliferation and differentiation [45]. The inconsistent effects of NGR1 treatment on different cell types of mice, rats, and humans may be attributed to the differential tolerance of NGR1 in different cell types. The lower concentration of drugs used in vivo ensures lower adverse effects. Therefore, in this study, we tested the effect of 0.01–5 µg/mL of NGR1 on the biological functions of hASCs and the results showed that 0.05 µg/mL of NGR1 was able to induce the cellular activities of hASCs required for bone tissue engineering application. Reports from literature had shown the different concentrations of NGR1 to promote the osteogenic differentiation of MSCs from different sources. Our results show that 0.05 µg/mL of NGR1 concentration can induce the osteogenic differentiation of hASCs. However, the exact molecular mechanism of precursor cell type-dependent optimal dose of NGR1 to induce osteogenic differentiation should be further studied.

Cell adhesion and spreading have been used as key parameters to evaluate the efficacy of surface compatibility of biomaterials for different types of cells in vitro [46,47]. The adhesion and spreading of cells onto the bone graft/implant surface are vital for successful stromal cell-based bone tissue engineering techniques [7]. Various bone grafts and implants with similar mechanical and physicochemical properties as bone fail osteointegration, is due to a lack of cell adhesive properties [48]. Bioactive molecules, such as collagen, fibronectin, RGD peptides, and basic fibroblast growth factor, have been widely used to improve the cell adhesive properties of biomaterials [49]. These bioactive molecules always required additional treatment to direct immobilization on the surface of the materials, which is often complex and time-consuming. Particularly, 3D-printed materials often need post-printing heating and sintering to get sufficient mechanical strength, which precludes the potential for simultaneous incorporation of heat-labile bioactive molecules. In this study, for the first time, we showed that 0.05 µg/mL of NGR1 optimally promoted the adhesion and spreading of hASCs on the glass surface and 3D-printed β-TCP scaffold surfaces.

The poor survival rate of MSCs during in vivo transplantation is one of the main drawbacks of the direct use of MSCs in bone tissue regeneration [50]. It has been reported that in vivo monitoring of transplanted MSCs in an acute myocardial infarction could identify only 4.4% of MSCs in the transplanted site after 1 week, which indicated the poor survival rate of transplanted MSCs [51]. In our study, we used the Live Image software with the IVIS Lumina LT Series III system to assess the survival rate of hASCs on 3D printed β-TCP scaffolds in vivo. At each time point, the survival rate of cells in the NGR1 group was higher than those in the control (1.65, 2.03, 1.70, and 6.22 times, respectively). In the NGR1 group, 10 days after injection with hASCs, 39.9% of cells in the implantation site were identified as hASCs, compared to 11.0% in the control group. Facilitating the migration of MSCs would increase the efficiency of MSC transplantation in clinical applications, resulting in significantly improved MSC-based cell therapy and regenerative medicine outcomes. Therefore, enhancing the migration of endogenous precursor cells in the defect site has become one of the prime aims of bone tissue engineering [52]. In this study, 0.05 and 5 μg/mL NGR1 robustly promoted the migration of hASCs.

The majority of bone grafts and implants have high osteoconductivity but lack osteogenic differentiation-inducing potential [53]. In bone tissue engineering, the incorporation of growth factors and various nanomaterials have been employed widely to drive the osteogenic differentiation of precursor cells. The shortcomings of these approaches, such as the low stability and high costs of growth factors [10,12], and the possible cytotoxicity of nanomaterials [54] often limit their clinic translation. In the past decade, engineered herbal constructs have received increasing attention owing to the biocompatibility, low toxicity, and cost-effectiveness of herbal extracts. In particular, some herbal extracts with pro-angiogenic and pro-osteogenic activities can add osteogenesis properties to biomaterials and increase the clinical efficacy of bone tissue engineering. For instance, icariin, isolated from several species of plants belonging to the genus *Epimedium*, was reported to have the potential to be applied in bone tissue engineering, owing to their biological functions, such as anti-osteoporotic, osteogenic, anti-osteoclastogenic, chondrogenic, angiogenic, and anti-inflammatory effects [55,56,57]. Ursolic acid is an active compound found in a variety of natural plants, which when loaded on biomaterial scaffolds has been shown to enhance bone regeneration by increasing the ALP activity and osteogenic differentiation-related protein and gene expression [58]. Otherwise, ursolic acid-loaded-mesoporous hydroxylapatite-chitosan scaffolds were reported to regulate bone regeneration ability by inhibiting the polarization of macrophages to M1 type [59]. In another study, grape seed, pomegranate peel, and jabuticaba peel extracts direct blend with scaffolds of nanohydroxyapatite and collagen showed potent anti-bacterial activity and offered a promising strategy to design novel biomaterials for bone tissue regeneration [60]. In addition, scaffolds containing herbal extracts, such as Cissus quadrangularis [61], kaempferol [62], aloe vera extract [63], and *Elaeagnus Angustifolia* extract [64], have shown a potential bone tissue regeneration ability. In addition, NGR1 significantly induces osteogenic differentiation of MC3T3-E1 cells in vitro [30]. In this study, NGR1 promoted the osteogenic differentiation of hASCs, as indicated by enhanced ALP activity, OCN expression, matrix mineralization, and osteogenic gene expression. Bone regeneration needs rapid vascularization to supply oxygen and nutrients. An increase in angiogenesis can lead to the restoration of damaged tissues, thereby leading the way for successful tissue regeneration. Our data has shown that NGR1 promoted the expression of proangiogenic growth factor VEGF.

The inflammatory response to chronic injury affects tissue regeneration and has become an important factor influencing the prognosis of patients [65]. It has been demonstrated that LPS-stimulated inflammation is significantly reduced by NGR1 in the atopic dermatitis model [66]. In this experiment, our data show that NGR1 had an anti-inflammation effect by effectively reducing IL-1β, IL-6, and TNF-α expression in LPS-treated hASCs. IL-1 induces bone destruction in a variety of diseases, such as osteoporosis, rheumatoid arthritis, and periodontal diseases [67]. IL-6 has been reported to increase osteoclastogenesis [68]. TNF-α is a common inflammatory cytokine elevated in chronic inflammatory conditions with poor bone healing [41]. IL-10 and TGF-β are important anti-inflammatory cytokines. NGR1 modulated mRNA expression of pro-inflammatory cytokines but not anti-inflammatory cytokines in LPS-treated hASCs. Reports from the literature showed the role of tumor necrosis factor receptor 1 (TNFR1) and tumor necrosis factor receptor 2 (TNFR2) in the immunomodulatory effect of MSCs [69,70,71]. The increased expression of TNFR1 is related to pro-inflammatory phenotypes and, inversely, the higher expression of the TNFR2 is related to anti-inflammatory phenotypes [70,72]. Therefore, the possible role of TNFR1/TNFR2 on NGR1-mediated anti-inflammatory effect on hASCs should be further explored. It is generally known that RANKL plays a key role in osteoclastogenesis and bone absorption. OPG is a critical negative modulator of bone resorption, which induces bone remodeling by regulating the RANKL/OPG ratio [67]. A previous study had shown that NGR1 significantly induces bone development by inhibiting RNAKL-mediated MAPK and NF-κB signaling pathways and suppressing osteoclastogenesis and bone resorption [73]. Our data showed that NGR1 mitigated the ratio of RANKL/OPG, which increased by LPS. Downregulating the ratio of RANKL/OPG in the bone microenvironment inhibits the differentiation and maturation of osteoclasts. Our results indicate that NGR1 has the potential to regulate the immune microenvironment during tissue repair. Further studies, such as the LPS-induced osteolysis model, are warranted to validate these inferences in vivo. Based on our results, in vitro NGR1 pretreated hASCs or a low dose NGR1-coating of 3D scaffold could improve the efficacy of bone tissue engineering. However, the in vivo bone regenerative potential of NGR1 with these approaches still needs to be further investigated in relevant experimental setups, such as critical-size bone defects reconstruction using 3D-printed scaffold and stromal cells. 

NGR1 is a phytoestrogen and binds to estrogen receptors of rat primary osteoblasts to promote osteogenic differentiation [45]. This suggests the estrogen signaling activation potential of NGR1 in osteoblast lineage cells. Further study is obligatory to unravel whether the estrogen signaling is involved in NGR1-mediated osteogenic differentiation of hASCs. Furthermore, the use of hASCs from a single donor is another limitation of this study. A future study using hASCs from multiple donors is recommended to corroborate the findings of the current study.

## 4. Materials and Methods

### 4.1. Cell Culture and Chemicals

The hASCs, purchased from Cyagen Biosciences Technology (Guangzhou, China), were isolated from the adult adipose tissue and cultured in hASCs complete growth medium (HUXMD-90011, Cyagen, Guangzhou, China). The standard characterization assay, including osteogenic, adipogenic, and chondrogenic differentiation assay, was performed by Cyagen as quality control. Cells from exponentially growing cultures at passages 4–5 were used for in vitro and in vivo experiments. NGR1 (C47H80O18, product number SN8230, purity ≥ 98%) was purchased from Solarbio^®^ Life Sciences (Beijing, China).

### 4.2. Cell Viability Assay

The hASCs (3000 cells/well) were seeded into a 96-well plate and treated with NGR1 (0, 0.01, 0.05, 0.5, and 5 µg/mL) for 1, 4 and 7 days. Cells were washed with PBS and incubated with a 110 µL fresh medium containing a 10 µL Cell Counting Kit (CCK)-8 solution (Dojindo Corp., Japan) for 4 h. Cell viability was determined by measurement of the absorbance using a spectrophotometer at a wavelength of 450 nm.

### 4.3. Alkaline Phosphatase (ALP) Staining and Activity Assay

For osteogenic differentiation studies, the osteogenic induction medium (hASCs complete growth medium with 50 μg/mL L-ascorbic acid (Sigma Aldrich, St. Louis, MO, USA), 10 mM β-glycerophosphate (Sigma Aldrich, St. Louis, MO, USA), and 10 nmol/L dexamethasone (Sigma Aldrich, St. Louis, MO, USA)) was added in the cultures. The hASCs (2.5 × 10^4^ cells/well) were seeded in 48-well plates. ALP activity was determined on days 4 and 7 using an ALP kit according to the manufacturer’s protocol (Nanjing Jiancheng Bioengineering Institute, Nanjing, China), and normalized to total protein content. Total protein was measured by a commercial BCA protein assay kit (Beyotime Institute of Biotechnology, Shanghai, China). The ALP staining was performed using a BCIP/NBT alkaline phosphatase color development kit (Beyotime Institute of Biotechnology, Shanghai, China) according to the manufacturer’s instructions.

### 4.4. Osteocalcin (OCN) ELISA

OCN protein release in the culture medium was measured on day 14. The supernatants of the cells were collected and OCN concentrations were detected with a human OCN enzyme-linked immunosorbent assay kit (SEA471Hu, Cloud-Clone Corp, Wuhan, China), according to the manufacturer’s instructions.

### 4.5. Alizarin Red Staining

Cells cultured in 48-well plates were stained with 2% alizarin red staining solution (pH 4.2) on days 21 and 28 to visualize the mineralized matrix in the hASCs culture. Images were obtained with a stereomicroscope (Leica, Singapore). To quantify the mineralized matrix, the alizarin red-stained calcium deposition was extracted with 10% cetylpyridinium chloride (CPC, Sigma Aldrich, St. Louis, MO, USA) for 20 min, and the absorbance of the extract was measured at 562 nm wavelength in a microplate reader.

### 4.6. The induction of the Inflammatory Microenvironment in hASCs 

Lipopolysaccharide (LPS) is a pro-inflammatory component of the cell membrane of Gram-negative bacteria, which could create an inflammatory microenvironment. The hASCs were exposed to 1 μg/mL LPS (*E. coli*) (Sigma Aldrich, Shanghai, China) for 4 h, washed twice with PBS, then, treated with different concentrations of NGR1 (0, 0.05, and 5 μg/mL) for 3 days.

### 4.7. RT-qPCR Analysis 

NGR1 or LPS-induced changes in gene expression of hASCs were analyzed using RT--qPCR. Total RNA was extracted using a SteadyPure Universal RNA Extraction Kit (Accurate Biology, Changsha, China) and reversed transcribed into cDNA using a PrimeScript RT reagent kit with gDNA Eraser (Takara, Dalian, China) according to the manufacturer’s protocols. The expression of ALP, COL1A1, OCN, VEGF, RANKL, OPG, IL-1β, IL-6, TNF-α, IL-10, and TGF-β were analyzed by RT-qPCR. The RT-qPCR was performed using the TB Green Premix Ex Taq II kit (Takara, Dalian, China). The conditions for PCR reaction were 1 cycle of 95 °C for 30 s, followed by 40 cycles of 95 °C for 5 s and 60 °C for 30 s. Each reaction was performed in triplicate. The 2^−ΔΔCT^ method was used to calculate the relative expression mRNA levels. The relative mRNA expression levels were standardized to the levels of the reference gene GAPDH. The primer sequences for the tested genes are listed in Table 1.

### 4.8. Cell Migration Assay

For the transwell assay, hASCs pretreated with NGR1 for 24 h (10,000 cells/well) suspended in serum-free medium were added into the upper chamber of a 24-well plate transwell insert. The serum-containing medium was placed in the lower chamber. After 16 h of incubation, the cells in the upper chambers were fixed with 4% paraformaldehyde for 20 min and stained with 0.1% crystal violet. The remaining cells on the inner side of the upper chamber were removed, and the migrated cells were imaged with a light microscope (Leica DMi1, Shanghai, China). The numbers of migrated cells were quantified using ImageJ. For migration analysis, cell numbers of five randomly selected fields per sample were measured for three samples per condition.

For the scratch wound assay, hASCs (4 × 10^5^ cells/well) were plated into 6-well plates 24 h prior to scratching. A 200 μL pipette tip was used to scratch the cell monolayer. Then, the cells were cultured in a low-serum medium (2% FBS) in the absence or presence of NGR1. Images were captured using a light microscope (Leica DMi1, Shanghai, China) at 0 h, 6 h. 12 h, and 18 h. The scratch area was assessed using ImageJ. Wound closure (%) = (original wound area − wound area at the metering point)/original wound area × 100.

### 4.9. Analysis of hASCs Adhesion and Spreading

The hASCs (1.2 × 10^4^ cells/coverslip) were seeded on glass coverslips (8 mm diameter) plated in a 48-well plate and treated with or without 0.05 µg/mL NGR1 for 1.5, 4, and 24 h. Cells were fixed with 4% fresh paraformaldehyde for 10 min at room temperature, permeabilized with 0.1% TritonX-100 in PBS for 10 min. Cells were incubated with 1% bovine serum albumin (BSA) in PBS at room temperature for 1 h to prevent non-specific binding. Subsequently, cells were stained with FITC-Phalloidin for 2 h at room temperature in the dark. After washing, cells were counterstained with DAPI for 5 min. After this, images were obtained with a 63× objective lens of Leica TCS SP8 confocal microscope (Leica, Germany). The attached number and surface area of hASCs were counted using the IN Cell Analyzer 2500HS (GE Healthcare, Issaquah, WA, USA). The attached number of cells was considered to represent cell adhesion. The surface area of cells was considered to quantitatively represent cell spreading [74,75].

We further analyzed the adhesion of NGR1 pretreated hASCs on 3D-printed β-tricalcium phosphate (TCP) scaffolds. The 3D TCP scaffolds were printed as follows: Briefly, β-TCP (Kunshan Chinese Technology New Materials Co., Ltd. Qingdao, China) ink was prepared by dispersing the powder in dispersant with distilled water, a proper amount of hydroxypropyl methylcellulose, and polyethylenimine (PEI) to increase agglomeration. A bio 3D printer (Regenovo, Hangzhou, China) was employed for the scaffold fabrication. The printing parameters were set at 700 μm in distance and 200 μm in height to obtain the final scaffold with a diameter of 5 mm and a thickness of 1 mm. After printing, the scaffolds were dried in air for 24 h and sintered at 1100 °C for 3 h. To seed the cells in the scaffolds, the sterile 3D TCP scaffolds were soaked in a culture medium overnight and dried in air. TPC scaffold was placed in a well of 48-well culture plates and covered with culture medium with or without 0.05 µg/mL NGR1. Cells (1 × 10^5^ cells/well) were dropped vertically on the TCP-plated wells and incubated for 1.5, 4, and 24 h. Subsequently, cells on scaffolds were fixed and stained. The images were captured with a Leica TCS SP8 laser scanning confocal microscopy. Image J software was used for cell counting.

### 4.10. Animal Study

All animal experiments obtained approval from the Animal Care Committee of PLA General Hospital of Southern Theatre Command, Guangzhou, China. PKH26-labeled hASCs were seeded on 3D-printed β-TCP scaffolds and treated with or without 0.05 µg/mL NGR1 for 3 h. Male nude mice (6 weeks old, 18–20 mg bodyweight) were anesthetized by intraperitoneal injection of 1% pentobarbital. Cell-scaffold constructs were carefully implanted in each of the two dorsal subcutaneous pockets. After implantation, the skin was closed using black non-resorbable 5-0 Mersilk sutures (Ethicon, Shanghai, China). Fluorescent images were acquired by the Living Image software with the IVIS Lumina LT Series III system (PerkinElmer, Hopkinton, MA, USA). The autofluorescence background signal intensity was subtracted. The fluorescence intensity was represented by a multicolor scale ranging from red (least intense) to yellow (most intense). Signal intensity images were superimposed over grayscale reference photographs for anatomical representations. In all experiments, signals were collected from a defined ROI using the contour ROI tool, and total radiant efficiency ([p/s]/[μM/cm^2^]) was analyzed using the Living Image software, 4.5.

### 4.11. Statistical Analysis

All quantitative data in this study represent the mean values ± SD (standard deviation) for n ≥ 3 (number of experiments). Data were analyzed using GraphPad Prism (GraphPad Software version 9.1, La Jolla, CA, USA). Significance differences were determined by one-way analysis of variance (ANOVA) with Bonferroni’s multiple comparison test, two-way ANOVA with Tukey’s multiple comparisons test, or the unpaired *t*-test. A value of *p* < 0.05 was considered statistically significant.

## 5. Conclusions

In summary, NGR1 enhances adhesion, spreading, migration, immunomodulation, and osteogenic differentiation of hASCs. In addition, NGR1 also enhanced the survival rate of hASCs on cell-scaffold constructs in vivo. This study explored the influence of NGR1 on hASCs’ function from different aspects required for effective bone regeneration.

## Figures and Tables

**Figure 1 molecules-27-03403-f001:**
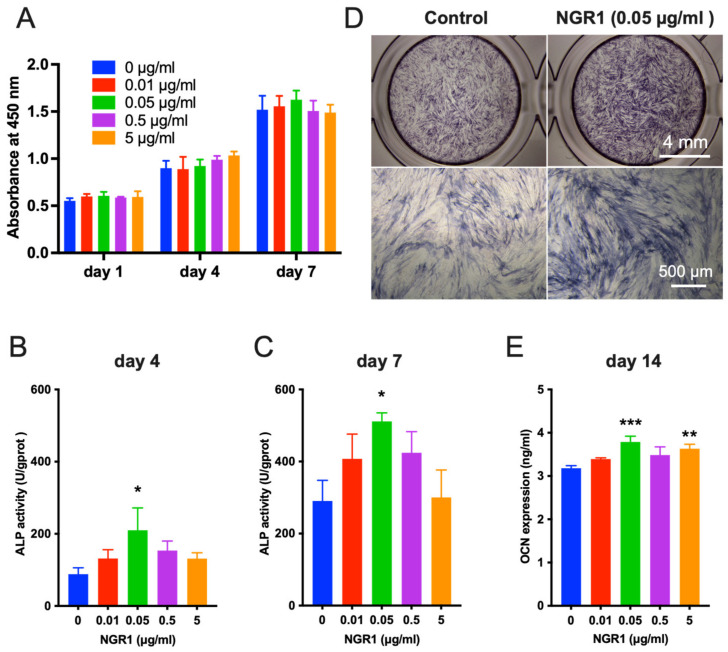
Biocompatible NGR1 promoted osteogenic differentiation of hASCs. (**A**) The proliferation of the hASCs under the different treatments with various concentrations of NGR1 for 1, 4, and 7 days. Cell viability was measured using a CCK-8 assay, n = 3. Data were analyzed with two-way ANOVA with Tukey’s multiple comparison test. (**B**) The activity of ALP in hASCs induced by NGR1 for 4 days, n = 3. (**C**) The activity of ALP in hASCs induced by NGR1 for 7 days, n = 3. (**D**) Cells were cultured in the absence or presence of 0.05 μg/mL NGR1. On day 4, cells were subjected to ALP staining. (**E**) Expression of osteocalcin (OCN) in hASCs under the different concentration NGR1 treatment for 14 days, n = 3. Data were analyzed with one-way ANOVA with Bonferroni’s multiple comparison test. * *p* < 0.05, ** *p* < 0.01, *** *p* < 0.001 relative to controls.

**Figure 2 molecules-27-03403-f002:**
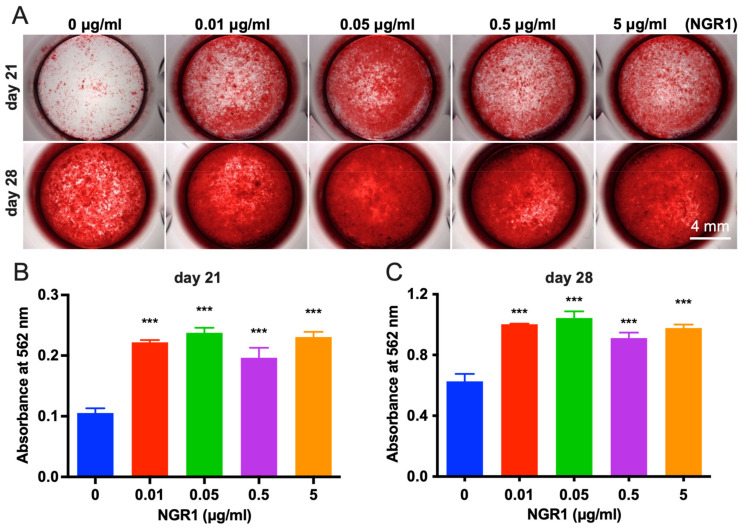
NGR1 promoted matrix mineralization in hASCs culture. (**A**) Light micrographs depicting alizarin red staining on days 21 and 28 of culture. (**B**,**C**) Quantitative analysis of mineralized matrix in hASCs culture, n = 3. Data were analyzed with one-way ANOVA with Bonferroni’s multiple comparison test. *** *p* < 0.001 relative to controls.

**Figure 3 molecules-27-03403-f003:**
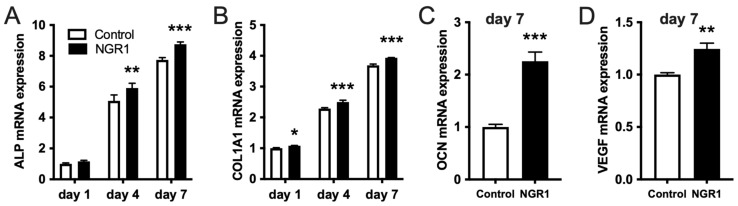
NGR1 upregulated the expression of osteogenic markers ALP, COL1A1, and OCN, as well as pro-angiogenic growth factor VEGF in hASCs. (**A**) ALP and (**B**) COL1A1 expression in hASCs treated with vehicle (control) or 0.05 µg/mL NGR1 for 1, 4 and 7 days, n = 3. Data were analyzed with two-way ANOVA with Tukey’s multiple comparison test. (**C**) OCN and (**D**) VEGF expression in hASCs treated with vehicle (control) or 0.05 µg/mL NGR1 for 7 days, n = 3. Data were analyzed with unpaired *t*-test.* *p* < 0.05, ** *p* < 0.01, *** *p* < 0.001 relative to controls.

**Figure 4 molecules-27-03403-f004:**
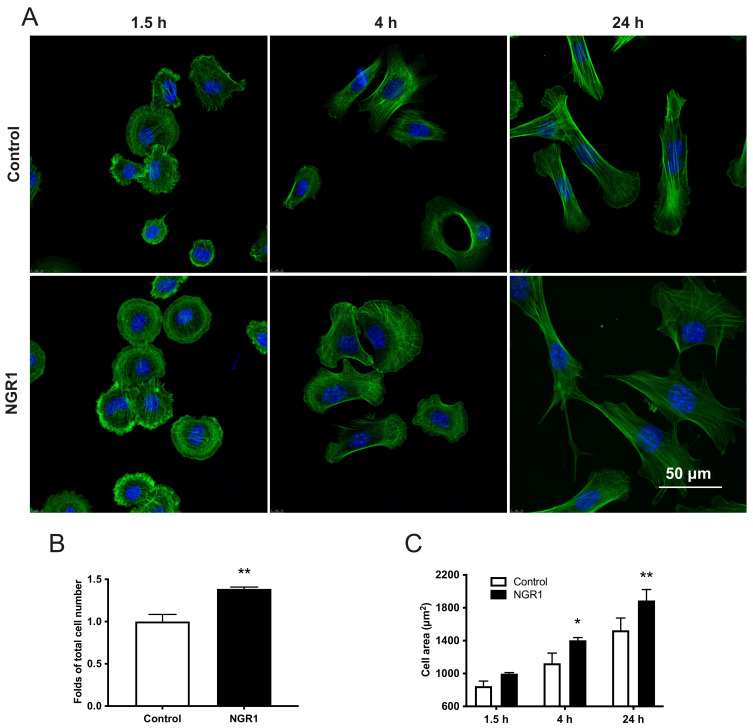
NGR1 promoted the adhesion and spreading of hASCs. (**A**) Images of hASCs on the glass surface. Green shows actin and blue shows nuclei. (**B**) Measurement of cell number after 1.5 h incubation, n = 3. Data were analyzed with unpaired *t*-test. (**C**) Quantification of hASCs spreading on the glass surface at 1.5, 4, and 24 h, n = 3. Data were analyzed with two-way ANOVA with Tukey’s multiple comparison test. * *p* < 0.05, ** *p* < 0.01 relative to controls.

**Figure 5 molecules-27-03403-f005:**
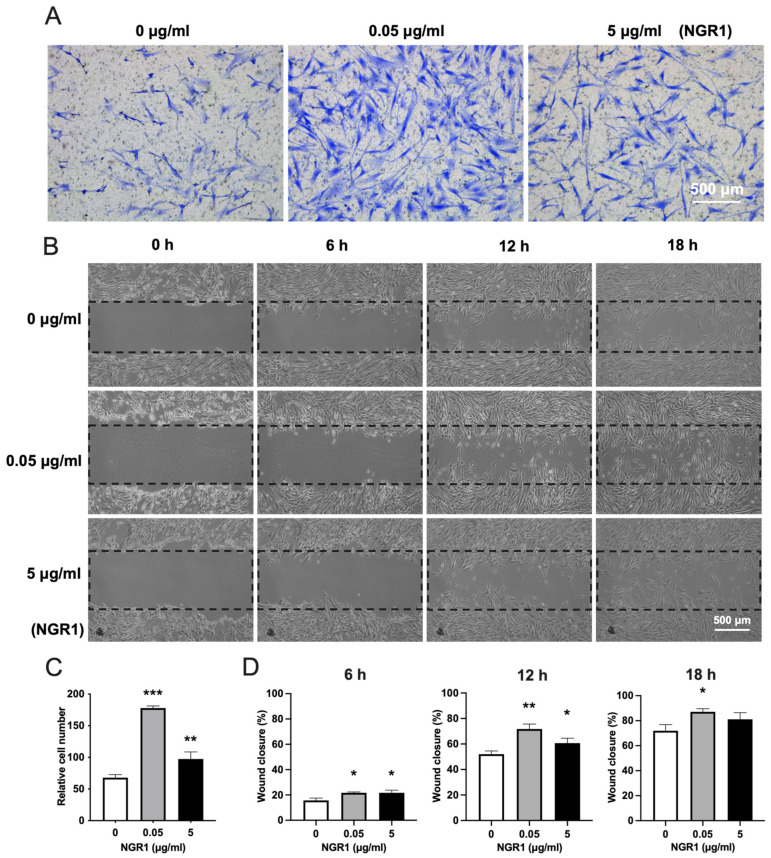
NGR1 promoted the migration of hASCs. (**A**) Representative images of transwell migration assay. (**B**) Representative images of scratch wound assay. (**C**) Quantitative data of migrated cells in transwell assay, n = 3. (**D**) Quantification of wound closure, n = 3. Data were analyzed with one-way ANOVA with Bonferroni’s multiple comparison test. * *p* < 0.05, ** *p* < 0.01, *** *p* < 0.001 relative to controls.

**Figure 6 molecules-27-03403-f006:**
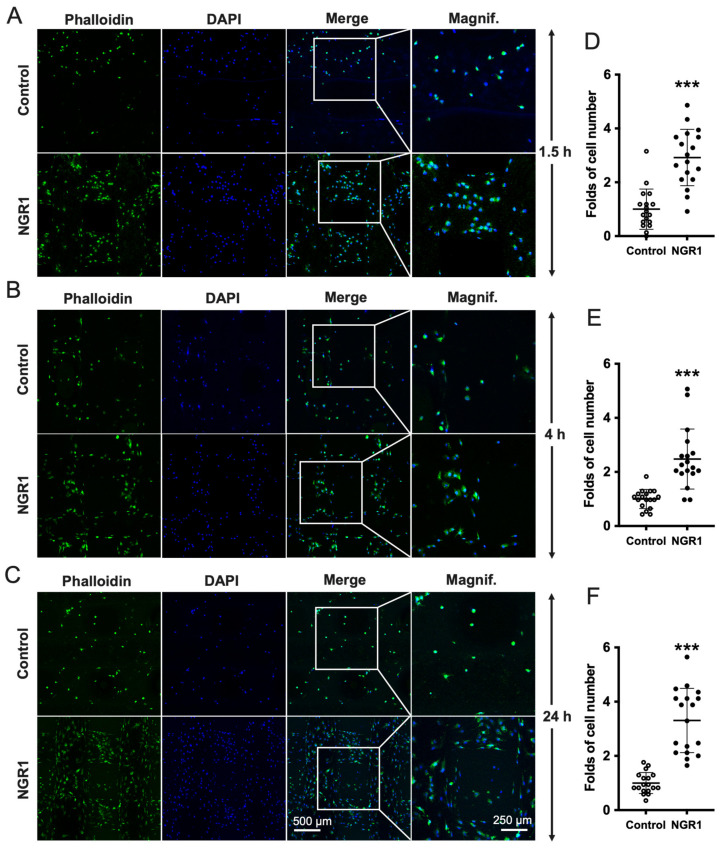
NGR1 enhanced hASCs adhesion on 3D-printed β-TCP scaffolds surface. Cells seeded on β-TCP scaffolds surface treated with or without NGR1 (0.05 μg/mL) for (**A**) 1.5 h, (**B**) 4 h, and (**C**) 24 h. Green shows actin and blue shows nuclei. The attached number of cells on β-TCP scaffolds surface (**D**) at 1.5 h, (**E**) at 4 h, and (**F**) at 24 h, n = 3. Data were analyzed with unpaired *t*-test. *** *p* < 0.001. Magnification: 2×.

**Figure 7 molecules-27-03403-f007:**
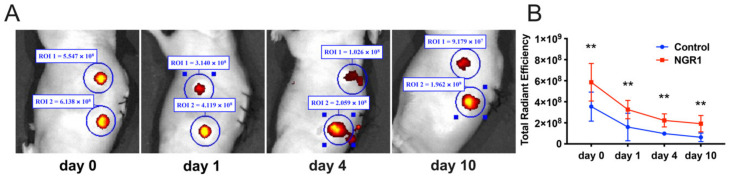
NGR1 enhanced the hASCs survival on 3D printed β-TCP scaffolds ectopically implanted in nude mice. (**A**) Representative images of cell-scaffold constructs implanted subcutaneously in nude mice. (**B**) Quantitative results of cells’ fluorescence intensity. Total radiant efficiency, n = 4. Data were analyzed with two-way ANOVA with Tukey’s multiple comparison test. ** *p* < 0.01 relative to control.

**Figure 8 molecules-27-03403-f008:**
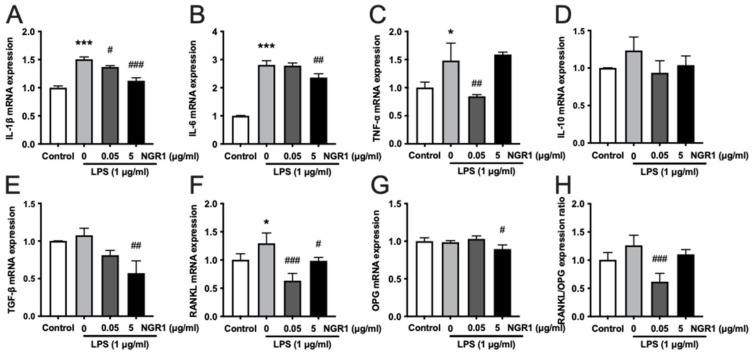
NGR1 showed anti-inflammatory properties in LPS-treated hASCs. Relative expression of (**A**) IL-1β, (**B**) IL-6, (**C**) TNF-α, (**D**) IL-10, (**E**) TGF-β, (**F**) RANKL, (**G**) OPG, and (**H**) RANKL/OPG in the inflammatory microenvironment, n = 3. Data were analyzed with one-way ANOVA with Bonferroni’s multiple comparison test. * *p* < 0.05, *** *p* < 0.001 relative to controls. # *p* < 0.05, ## *p* < 0.01, ### *p* < 0.001 relative to LPS (without NGR1) group.

**Figure 9 molecules-27-03403-f009:**
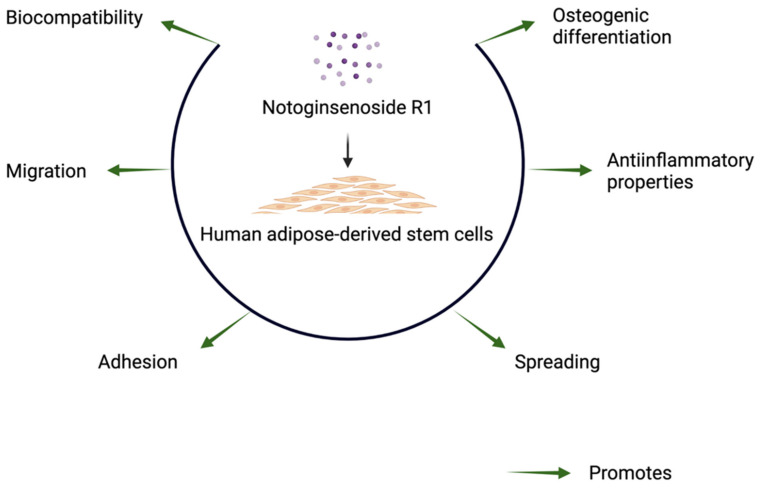
NGR1 promotes adhesion, spreading, migration, immunomodulation, and osteogenic differentiation potential of hASCs. The images were created with BioRender.com accessed on 3 March 2022.

**Table 1 molecules-27-03403-t001:** Primers used for RT-qPCR analysis.

Gene	Primer Sequences
ALP	Forward: 5′-GGACCATTCCCACGTCTTCAC-3′ Reverse: 5′-CCTTGTAGCCAGGCCCATTG-3′
COL1A1	Forward: 5′-GTGCCAAGGGTCTGACTGGAA-3′ Reverse: 5′-ATCACACCAGCCTGACCACG-3′
OCN	Forward: 5′-CTCACACTCCTCGCCCTATTGG-3′ Reverse: 5′-GTAGCGCCTGGGTCTCTTCACT-3′
VEGF	Forward: 5′-GGAGGCAGAGAAAAGAGAAAGTGT-3′ Reverse: 5′-TAAGAGAGCAAGAGAGAGCAAAAGA-3′
RANKL	Forward: 5′- TGATGAAAGGAGGAAGCA-3′ Reverse: 5′- GTAAGGAGGGGTTGGAGA-3′
OPG	Forward: 5′-AACCCCAGAGCGAAATAC-3′ Reverse: 5′-AGCAGGAGACCAAAGACAC-3′
IL-1β	Forward: 5′-ATGATGGCTTATTACAGTGGCAA-3′ Reverse: 5′-GTCGGAGATTCGTAGCTGGA-3′
IL-6	Forward: 5′-ACTCACCTCTTCAGAACGAATTG-3′ Reverse: 5′-CCATCTTTGGAAGGTTCAGGTTG-3′
TNF-α	Forward: 5′-GAGGCCAAGCCCTGGTATG-3′ Reverse: 5′-CGGGCCGATTGATCTCAGC-3′
IL-10	Forward: 5′-GACTTTAAGGGTTACCTGGGTTG-3′ Reverse: 5′-TCACATGCGCCTTGATGTCTG-3′
TGF-β	Forward: 5′-CAATTCCTGGCGATACCTCAG-3′ Reverse: 5′-GCACAACTCCGGTGACATCAA-3′
GAPDH	Forward: 5′-GCACCGTCAAGGCTGAGAAC-3′ Reverse: 5′-TGGTGAAGACGCCAGTGGA-3′

## Data Availability

The data presented in this study are openly available in Figshare at 10.6084/m9.figshare.19375136.v1 accessed on 17 March 2022.

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
