# Peer review of "Notoginsenoside R1 Promotes Migration, Adhesin, Spreading, and Osteogenic Differentiation of Human Adipose Tissue-Derived Mesenchymal Stromal Cells"

_molecules, 2022, doi:10.3390/molecules27113403_

Round 1
Reviewer 1 Report
The effects of Notoginsenoside R1 (NR1), a bioactive compound extracted from Panax notoginseng, on bone tissue repair were evaluated. The properties of NR1 were firstly tested in vitro and subsequently confirmed in vivo. For the in vitro studies, a human adipose-derived stem cells lineage (hASCs) was employed to show the NR1 effects on cell adhesion, proliferation and migration, osteogenic differentiation, expression of pro-inflammatory molecules and the growth factor VEGF. Data presented are very interesting and promising. However, some points should be addressed.
Major revisions:
The major concern, in my opinion, is that the data are not truly discussed. The results are repeated at Discussion section and discussion seems fragmentary.
For instance, there was no stated how and why concentrations of NR1 have been choose.
There is no discussion why concentration of 0.05 ug/ml presented higher effects on ALP activity and OLC secretion.
How NR1 displays the effects? Is there any receptor where NR1 act in/on the cell? Has it been described? It should be discussed.
Furthermore, it should be discussed how data obtained in vitro could corroborate data in vivo. Why was the scaffold implanted? Was it expected that those effects observed in vitro could contributed to the effects observed in vivo? Was the implanted local collected to evaluate the tissue histology? Was blood of mice collected? Could be possible to evaluate levels of systemic cytokines as IL-6, IL-1beta and VEGF. Was evaluated if the implanted material presented any toxicity (local or systemic)?
Minor revisions:
- Introduction: Although some data regarding the prevalence of bone diseases in population have been mentioned in the Introduction, I missed epidemiological data. Additionally, data presented in the manuscript seems very interesting, but it is not emphasized. Introduction seems confused and insufficient organized. Moreover, the Introduction could be finalized with the main results obtained in the work and why data obtained could contribute to the bioengineering. In my opinion, ending the introduction in this way draws the reader's attention to continue reading the manuscript.
Author Response
Comment 1. The major concern, in my opinion, is that the data are not truly discussed. The results are repeated at Discussion section and discussion seems fragmentary. For instance, there was no stated how and why concentrations of NR1 have been choose.
Reply: We rewrote the discussion section as suggested by the reviewer. We added the following text in the discussion section (page 11, line 313-325): “NGR1 promotes osteoblast differentiation, but the optimal concentration of NGR1 treatment remains debatable. Liu et al. reported that treatment with 5-1000 µg/ml NGR1 promoted osteogenic differentiation of MC3T3-E1, while NGR1 at 200 and 1000 µg/ml significantly inhibited cell proliferation [30]. Huang et al. reported that NGR1 at a concentration of ≤ 20 µM could promote differentiation of human alveolar osteoblasts (HAOBs) in TNF-α induced inflammatory microenvironment [44]. NGR1 at a concentration of ≤ 20 µM stimulates rat osteoblast proliferation and differentiation [45]. The inconsistent effects of NGR1 treatment on different cell types of mice, rats, and humans may be attributed to the differential tolerance of NGR1 in different cell types. The lower concentration of drugs used in vivo ensures lower adverse effects. Therefore, in this study we tested the effect of 0.01-5 µg/ml of NGR1 on the biological functions of hASCs and the results showed that 0.05 µg/ml of NGR1 was able to induce cellular activities of hASCs required for bone tissue engineering application.”
Comment 2. There is no discussion why concentration of 0.05 ug/ml presented higher effects on ALP activity and OLC secretion.
Reply: We added the following text in the discussion section (page 11, line 325-330): “Reports from literature had shown the different concentrations of NGR1 to promote the osteogenic differentiation of MSCs from different sources. Our results show that 0.05 µg/ml of NGR1 concentration can induce the osteogenic differentiation of hASCs. However, the exact molecular mechanism of precursor cell type-dependent optimal dose of NGR1 to induce osteogenic differentiation should be further studied.”
Comment 3. How NGR1 displays the effects? Is there any receptor where NR1 act in/on the cell? Has it been described? It should be discussed.
Reply: We added the following text to the discussion section of the manuscript (page 13, line 429-433): “NGR1 is a phytoestrogen and binds to estrogen receptors of rat primary osteoblasts to promote osteogenic differentiation [45]. This suggests the estrogen signaling activation potential of NGR1 in osteoblast-lineage cells. Further study is obligatory to unravel whether the estrogen signaling is involved in NGR1-mediated osteogenic differentiation of hASCs.”
Comment 4. Furthermore, it should be discussed how data obtained in vitro could corroborate data in vivo. Why was the scaffold implanted? Was it expected that those effects observed in vitro could contributed to the effects observed in vivo? Was the implanted local collected to evaluate the tissue histology? Was blood of mice collected? Could be possible to evaluate levels of systemic cytokines as IL-6, IL-1beta and VEGF. Was evaluated if the implanted material presented any toxicity (local or systemic)?
Reply: The main aim of the in vivo study was to analyze in vivo viability of hASCs seeded on bone grafts. Therefore, we only perform in vivo imaging and did not perform histology and cytokines level analysis in serum, and the toxicity of implanted materials. However, our ongoing experiment is mainly focused on these studies, including in vivo bone regenerative potential. For better clarification we added the following text in the discussion section (page 13, line 423-428): “Our results indicate that NGR1 has the potential to regulate the immune microenvironment during tissue repair. Further studies, such as the LPS-induced osteolysis model are warranted to validate these inferences in-vivo. Moreover, the in-vivo bone regenerative potential of NGR1 still needs to be further investigated in relevant experimental set-ups, such as critical-size bone defects reconstruction using 3D-printed scaffold and stromal cells.”
Comment 5. Introduction: Although some data regarding the prevalence of bone diseases in population have been mentioned in the Introduction, I missed epidemiological data. Additionally, data presented in the manuscript seems very interesting, but it is not emphasized. Introduction seems confused and insufficient organized. Moreover, the Introduction could be finalized with the main results obtained in the work and why data obtained could contribute to the bioengineering. In my opinion, ending the introduction in this way draws the reader's attention to continue reading the manuscript.
Reply: We added the following text in the Introduction section of the manuscript (page 2, line 55-59): “Osteoporotic fractures are extremely common in the United States, with an estimated 1.5 million fragility fractures each year. In the UK, one in two women and one in five men over the age of 50 years may suffer an osteoporotic fracture in their lifetime [4]. More than 2 million bone defect repair surgery are performed worldwide annually [5].”
(page 3, line 110-113): “Our results indicated that NGR1 at a concentration of 0.05 µg/ml is biocompatible to hASCs and promotes migration, adhesion, spreading, osteogenic differentiation, and anti-inflammatory properties of hASCs suggesting the potential application of NGR1 in bone tissue engineering.”

Reviewer 2 Report
The study by H. Wang and colleagues investigates the effect of NGR on osteogenesis of adipose tissue -derived stem cells. They report that NGR1 facilitated osteogenic differentiation of ADSCs in vitro and at the same time mitigated inflammatory processes. The study investigated the biological effects on several distinct levels including enzymatic activities associate with bone repair, extracellular matrix deposition and mineralization, transcript expression, cell attachment and others. Most important, the study includes an implantation study s.c. in immunodeficient mice to document differences in cell survival after NGR treatment of ADSCs in a pre-clinical situation.The different experiments and analyses were performed and are presented well.
Critiques:
The study has bee performed only with 1 batch of cells. This means that the statistics reported document the precision of performance of that very experiment ( e.g., pipetting, counting, ...). Conclusions on the clinical applicability are therefore not well supported by the experiments performed and statistics can't give any evidence on the cell responses of ADSCs in general. To make conclusions on the clinical potential, cells from relevant cohorts and sufficient numbers of individuals must be included.
For instance: In figures 1E, 2B-day 1, -day 4, and -day7 statistically significant differences are reported. I wonder how such a minor difference ran reach so high ( ***!) significance when only 3 analyses wer performed with 1(!) batch of (commercially available) ADSCs.
Minor:
some very minor flaws in language should be addressed
the term "stem" cell in MSCs, ADSCs, etc should be placed by "stromal" cell based on consensus papers published a decade ago
check expression in lines 63 - 65
line 421 in the conclusions the author should limit their statements to ADSCs
lines 437/438: The information on approval of the animal study lacks a file number and date.
Author Response
Comment 1. The study has been performed only with 1 batch of cells. This means that the statistics reported document the precision of performance of that very experiment ( e.g., pipetting, counting, ...). Conclusions on the clinical applicability are therefore not well supported by the experiments performed and statistics can't give any evidence on the cell responses of ADSCs in general. To make conclusions on the clinical potential, cells from relevant cohorts and sufficient numbers of individuals must be included.
Reply: We sincerely thank the reviewer for the pertinent question. We indeed use only one batch of cells, and this is a limitation of this study. We highlighted the limitations in the Discussion section as follows (page 13, line 433-435): “Furthermore, the use of hASCs from a single donor is another limitation of this study. A future study using hASCs from multiple donors is recommended to corroborate the findings in the current study.”
Comment 2. For instance: In figures 1E, 2B-day 1, -day 4, and -day7 statistically significant differences are reported. I wonder how such a minor difference ran reach so high ( ***!) significance when only 3 analyses were performed with 1(!) batch of (commercially available) ADSCs.
Reply: We reconfirmed this high significance with minor differences was mainly due to a small standard deviation. In figure 1E, we used one-way ANOVA with Bonferroni’s multiple comparison test to analyze the differences and in figure 3B, we used two-way ANOVA with Tukey’s multiple comparisons test for statistical analysis.
Comment 3. Some very minor flaws in language should be addressed
Reply: We tried to correct the minor flaws throughout the manuscript. The changes are highlighted.
Comment 4. the term "stem" cell in MSCs, ADSCs, etc should be placed by "stromal" cell based on consensus papers published a decade ago
Reply: We made the changes accordingly throughout the manuscript.
Comment 5. check expression in lines 63 – 65
Reply: We rewrote the sentence as follows: "Natural small molecule compounds can trigger certain cellular responses through signaling cascades which can exert anti-tumor [14], anti-oxidant [15], anti-bacterial [16], anti-inflammatory [17], and pro-osteogenesis [18] effects."
Comment 6. line 421 in the conclusions the author should limit their statements to ADSCs
Reply: We made the changes accordingly.
Comment 7. lines 437/438: The information on approval of the animal study lacks a file number and date.
Reply: We added this information.

Reviewer 3 Report
This manuscript describes the impact of NGR1 priming on adipose-derived MSCs. The authors have treated MSCs with different doses of NGR1 and demonstrated that while treated with 0.05 ug/ml, MSCs have higher adhesion, migration, and osteogenic capacities. Moreover, they have shown that this dose of NGR1 can decrease the secretion of IL-1b, TNFa, and IFNg by MSCs.
Although this manuscript is interesting, the following points should be addressed:
* In order to better measure the migration and degenerative capabilities of MSCs, the authors should have performed the scratch test. (With and without NGR1)
* The authors have shown that 0.05 ug/ml of NGR1 has increased cell adhesion and migration. However, the time frame was too short. Did they test some longer time points such as 48h, 72h, and 96h?
* The authors should have tested the direct effect of different doses of NGR1 on MSCs. Moreover, they should have tested these cytokines at protein levels.
* Several recent articles demonstrate the role of the TNF/TNFR2 signaling pathway in MSC biology and survival. It was shown that the increased expression of TNFR1 is related to pro-inflammatory phenotypes and inversely, the higher expression of the TNFR2 is related to anti-inflammatory phenotypes. The authors are requested either to measure them on their MSC or to comment on that. PMIDs: 33344453, 33303019, 32669116, 34567420.
* Regarding anti-inflammatory cytokines no information is provided. What about IL-10 and TGFb secretion?
* The authors did not provide any information about MSC characteristic markers in the method section.
Round 2
Reviewer 2 Report
The study brings little progress to the field. For real clinical applications of cells ± factors such as NGR1 several draw-backs of the experimental design and especially hurdles in the process of approval as therapy use must be addressed. In this sense, many conclusions are not supported in a strict sense by the experiments performed. e.g., : 1) commercial hASCs were used. These cells are by far not produced under conditions compliant to any of the GMP-protocols published by Health Authorities of countries with responsible registries of medical procedures 2) the authors did not expand these cells employing protocols compliant with GMP-procedures. But it is state of the art that alterations in composition of cell culture media will have e.g., significant effects on the phenotype of a ASC, including hASCs. 3) Risk management of ASCs ( e.g. growth w/o antibiotics, detailed phenotype analyses, ... ) was not performed at al. 4) The authors provide no experimental evidence if GMP-compliant procedure of NGR1 is possible. Therefore the study lacks convincing evidence that NGR1 promotes activities (quote the title) "required for tissue engineering" . Moreover, they do not consequently discriminate if the intend to use NGR1 to complement in vivo implants or to add NGR1 for tissue engineering of in vitro implants. I would have much less concern if the authors would agree to delete all statements ( starting at the title to the conclusion. i.e. line 485 ) that give the reader the impression that these very results can be translated (directly) in clinical studies.E.g., NGR1 seems not REQUIRED for bone tissue engineering, it will only enhance it to some extent.
Author Response
We agree with the reviewer’s concern regarding the GMP-procedures for hASCs harvesting, expanding, and phenotyping before the clinical application. However, this study is still in too early stage for clinical application. Since, we just performed in vitro studies to analyze the effect of NGR1 on the basic functions of hASCs. Similarly, regarding the GMP-compliant procedure of NGR1, we did not isolate NGR1 ourselves, NGR1 (C47H80O18, product number SN8230, purity ≥ 98%) was purchased from Solarbio® Life Sciences (Beijing, China). Therefore, we did not perform the additional experiment to NGR1 related GMP-compliant procedures.
Regarding the possible in vivo application of NGR1, we added the following text in the Discussion section (page 13, line 340-344): “Based on our results, in vitro NGR1 pretreated hASCs or low dose NGR1-coating of 3D-scaffold could improve the efficacy of bone tissue engineering. However, the in-vivo bone regenerative potential of NGR1 with these approaches still needs to be further investigated in relevant experimental set-ups, such as critical-size bone defects reconstruction using 3D-printed scaffold and stromal cells.”
As suggested by the reviewer, we deleted the statement "required for tissue engineering" throughout the manuscript.
Reviewer 3 Report
The authors have successfully addressed most of my concerns. They have performed complementary experiments to clarify some points.
The manuscript is well ameliorated, therefore, I have no more comments.
Author Response
We thank the reviewer for the thorough revision and approval of this manuscript for publication.